# A Narrative Review of Specialist Parkinson’s Nurses: Evolution, Evidence and Expectation

**DOI:** 10.3390/geriatrics7020046

**Published:** 2022-04-07

**Authors:** Emma Tenison, Alice James, Louise Ebenezer, Emily J. Henderson

**Affiliations:** 1Department of Population Health Sciences, Bristol Medical School, University of Bristol, Bristol BS8 1NU, UK; aj1749@bristol.ac.uk (A.J.); emily.henderson@bristol.ac.uk (E.J.H.); 2Cwm Taf Morgannwg University Health Board, Princess of Wales Hospital, Coity Road, Bridgend CF31 1RQ, UK; louise.ebenezer@wales.nhs.uk; 3Older People’s Unit, Royal United Hospitals Bath NHS Foundation Trust, Combe Park, Bath BA1 3NG, UK

**Keywords:** Parkinson’s disease, nursing, effectiveness of care, caregivers

## Abstract

Extended nursing roles have existed since the 1940s. The first specialist nurse for Parkinson’s disease, a complex neurodegenerative disease, was appointed in the United Kingdom (UK) in 1989. A review was undertaken using MEDLINE and Cumulative Index to the Nursing and Allied Health Literature (CINAHL), relating to the role and evidence for Parkinson’s disease nurse specialists (PDNSs). PDNSs fulfil many roles. Trials of their effectiveness have failed to show a positive benefit on health outcomes, but their input appears to improve the wellbeing of people with Parkinson’s. Now embedded in the UK Parkinson’s multidisciplinary team, this care model has since been adopted widely, including successful dissemination of training to countries in Sub-Saharan Africa. The lack of evidence to support the benefit of PDNSs may reflect an insufficient duration and intensity of the intervention, the outcome measures selected or the need to combine PDNS input with other evidence-based interventions. Whilst the current evidence base for their effectiveness is limited, their input appears to improve subjective patient wellbeing and they are considered a vital resource in management. Better evidence in the future will support the development of these roles and may facilitate the application of specialist nurses to other disease areas.

## 1. Introduction

Specialist nursing roles are a core component of healthcare delivery [1]. In neurodegenerative disease, nurses with extended expertise in Parkinson’s are a critical member of multidisciplinary teams (MDTs) in the United Kingdom (UK) and, increasingly, the world over [2]. We look here at the origin of the role in the UK, over three decades ago, how the role has evolved and look at potential future avenues.

Extended nursing roles originated in 1943, when American nurse, Frances Reiter, coined the term ‘nurse clinician’. This described a nurse who had developed their knowledge and clinical competence beyond that of their ‘basic’ nursing training, but whilst still continuing ongoing clinical practice [3]. In the 1950s, a programme to train advanced practitioners in psychiatric nursing provided one opportunity to develop a specialty interest [4]. Since that time, specialist nurses, defined as those with “advanced expertise in a clinical speciality” [5], have become central to the care of individuals with several chronic diseases. There is some ambiguity around the roles and terminology for such advanced nursing positions, with variation between countries. In North America, the term ‘advanced practice nurse’ is used as an umbrella term for nurse practitioners and clinical nurse specialists (CNSs); in the UK, a CNS typically focuses on a particular subspeciality, whilst an advanced nurse practitioner (ANP) is often based in a more generalist area, such as the emergency department or general practice [6].

The nurse specialist role developed, partly, as a response to growing pressures on consultant-led services and the finding that patients value their input. In the early 1980s, a rheumatology nurse specialist role was developed at Leeds General Infirmary to meet demands on the service and noting patients in trials preferred the frequent, longer appointments with the research nurses, compared to the short, infrequent reviews by their physician [7]. Nurse-led care is now embedded within multiple disease areas, including diabetes [8], multiple sclerosis [9] and chronic rheumatological conditions [10]. 

Neurological diseases are the leading cause of disability globally [11]. Parkinson’s disease (PD) is the second most common neurodegenerative condition and affects over 10 million people worldwide [12,13]. Parkinson’s disease encompasses a spectrum of motor and non-motor symptoms that manifest across the disease course and fluctuate over time. Existing pharmacological therapies tend to focus on symptomatically improving motor symptoms. However, it is increasingly recognised that non-motor symptoms, including depression, cognitive impairment, urinary/faecal incontinence, pain, and dizziness are associated with reduced quality of life in PD [14,15]. The disconnect between physician and patient perspectives on the symptoms has been described in studies worldwide [16]. Use of a multidisciplinary team to exploit a range of expertise and support is one approach that will likely bridge the gap in care [16]. Acknowledging the psychological impact that neurological diseases, including PD, have on patients and their family, a Royal College of Physicians working group, writing in 1990, suggested a role for ‘keyworkers’ [17]. They suggested these keyworkers could be from a range of professional backgrounds but would need to develop cross-skilling so that they could provide initial advice, coordinate care and signpost on as needed [17].

The first UK-based Parkinson’s Disease Nurse Specialist (PDNS), Rosemary Maguire (Figure 1), was appointed in 1989. She went on to develop a course in ‘Specialist care of people with Parkinson’s’ at Plymouth University and was awarded an MBE (Member of the Most Excellent Order of the British Empire) for her contribution to Parkinson’s care [18,19]. A team of five PDNSs were set up by the Parkinson’s Disease Society in 1992 to improve standards of care for people with PD [20].

In this narrative review, we address (a) the role of a PDNS and expertise required, with a particular focus on the UK system, (b) the current evidence base of effectiveness, and (c) a perspective on dissemination of the role globally. We synthesise and critically appraise the literature in the field and consider the future horizons.

## 2. Methods

A review was undertaken of national and international literature using MEDLINE (on Ovid SP) and Cumulative Index to the Nursing and Allied Health Literature (CINAHL) to search for published literature dating from 1 January 1990 to 22 October 2021, relating to the role, global reach and evidence for Parkinson’s disease nurse specialists. Comprehensive electronic searches were conducted combining the following MeSH and text word search terms: “Parkinsonian disorders”, “Parkinson*”, “Parkinson disease”, “Nurses”, “Nurse specialists”, “nurse adj specialist”. The search was restricted to the English language; viewpoints were included as well as original articles. Further articles were obtained by reviewing reference lists. Relevant grey literature, such as policy documents, reports and websites of third sector organisations and professional societies, were also included. A narrative review was chosen, in preference to a systematic review, as we also sought to describe the emergence of PD nurse specialists and their implementation internationally, rather than solely focusing on the effectiveness of nurse intervention. When reviewing the evidence for effectiveness, we focused on studies where the primary or single focus was a nursing intervention rather than where a nursing intervention was part of a broader multidisciplinary intervention study.

## 3. Results

### 3.1. The Role of a Parkinson’s Disease Nurse Specialist

Given the heterogeneity of the condition, there is an increasingly recognised need to personalise PD management, taking into account an individual’s phenotype and symptoms [21] and to ensure good psychological and social support, alongside pharmacotherapy, in what has been described as a ‘biopsychosocial approach’ to PD [22]. With their roles encompassing diverse functions, summarised in Figure 2 and described below, PDNSs are critical to providing this wider support and to establishing and maintaining good-quality care in PD. It has been suggested that patients may be more prepared to discuss sensitive issues, such as sexual dysfunction, with their PDNS than with their physician [23].

One of the many roles fulfilled by PDNSs is to provide emotional and psychological support to patients and caregivers [2]. The need for emotional support and empathy was the most commonly identified theme resulting from focus groups exploring patient-centredness with people with Parkinson’s and their caregivers [27]. A diagnosis of PD can have a huge physical, psychosocial and financial impact on those close to the person with the condition [28]. Qualitative research has revealed that a top patient priority is for their spouse’s needs to be considered [29]. Dyad interviews with PD patients and their spouses, conducted in Sweden, highlighted the crucial role PDNSs play in supporting patients and caregivers to accept the change to their life, set goals and plan for future disease progression [30]. A PDNS has the skills to support individuals throughout all disease stages [25], including timely initiation of advance care planning discussions in anticipation of the complex and palliative phases [24].

A key role of the PDNS is to educate patients, caregivers, and non-specialist health and social care professionals [2]. Patients and caregivers wish to be able to access reliable information, tailored to their needs [27], along with a healthcare professional who can act as a single point of access, to provide information or signpost as appropriate [29]. In addition to providing practical advice and information, including about access to financial benefit [31], PDNSs have a central role in helping patients to develop skills to self-manage their disease, improve their lifestyle and actively participate in decision making [32]. It is increasingly recognised that supporting and empowering patients to look after their health has positive effects on health outcomes [28] and, as such, supporting a self-management approach is a key part of the UK Competency Framework for Parkinson’s nurses [24].

The PDNS uses their Parkinson’s-specific expertise to assess and monitor symptoms, overseeing and, in the case of nurse prescribers, facilitating often complex drug regimens [2]. They also have a role in identifying patients who may benefit from non-oral therapies: deep brain stimulation, apomorphine and levodopa-carbidopa intestinal gel [33,34]. Bhidayasiri et al. highlighted how the PDNS is involved at all stages of apomorphine therapy, contributing to its potential success [33]. The PDNS also has an important role in coordinating care and is often the main link between the patient, primary and secondary care [2,25]. People with PD have reported their healthcare to be fragmented, expressing a desire to get to know a particular professional over a prolonged period and for their healthcare professionals to collaborate together [29]. PDNSs can provide this continuity, as well as coordinating referrals to the wider MDT [30].

### 3.2. The Evidence for Effectiveness

We identified four RCTs, which compared models of care, involving PDNSs against those without PDNS input. These trials are summarised in Table 1 and described below.

Jahanshahi et al. conducted an RCT to assess the impact of contact with a nurse practitioner, via two home visits and five telephone calls, compared to usual care, amongst 40 patients with PD and 24 patients with dystonia [31]. The nurse practitioners delivering the intervention had three main roles: provision of information about the disease and approaches to self-management; assessing needs and facilitating referral to the multidisciplinary team; general support, including a point of contact between scheduled visits and calls. All patients completed psychosocial health questionnaires at the start and end. In addition, participants from the intervention group were contacted by investigators not involved in intervention delivery and invited to anonymously answer questions about their satisfaction with the nurse practitioner contact. There was no evidence of a difference in psychosocial wellbeing between the groups. However, the satisfaction questions, reported separately from the main analysis, found that participants valued nurse practitioner input and 96.2% (*n* = 25) agreed this should be an important priority for the health service [26]. Answering one free text question, a participant commented ‘The principal advantage lay in the contact with a qualified professional, away from the sometimes intimidating atmosphere of a large and busy hospital [31]’.

Reynolds et al. compared outpatient services led by a consultant neurologist only, PDNS only and combined PDNS/consultant follow-up in an RCT, with frequency of contact based on patient need [20]. Of the 185 patients enrolled, 58% (*n* = 108) of patients completed the study. The high dropout rate is said to be due to advancing illness, which led to participants being unable to understand the study requirements and attend clinics. The primary outcome was health state, assessed using six questionnaires, from which the authors derived 22 dimensions. The outcomes were similar between all groups, with the exception of the physical functioning and general health domains of the 36-item short form survey, which favoured consultant-led care. The authors concluded that, whilst the complementary expertise of PDNSs was valued, the increased cost meant they could not be recommended [20].

Jarman et al. conducted a two-year RCT, which assessed the impact of community-based PDNSs on health outcomes and healthcare costs. The nurses, who were new to the community-based PDNS role but received specialist training, monitored response to treatment, helped to educate patients and carers, liaised with the multidisciplinary team and advised general practitioners on potential medication changes. As such, 1859 eligible patients, sampled from general practices in nine English health authorities, were randomised within practices to either receive standard care from their general practitioner or to receive input from a PDNS in the community [35,37]. The randomisation ratio varied between 50:50 to 70:30, in order to standardise the nurses’ workload. Objective health measures and healthcare costs were assessed as summarised in Table 1. There was no evidence of a difference in the severity of PD symptoms between the intervention and control groups at 2 years and no difference in health-related quality of life, although a global health question indicated PDNS care improved subjective wellbeing [35].

Connor et al. assessed the impact of PDNS-led care, as part of the Care Coordination for Health Promotion and Activities in Parkinson’s Disease (CHAPS) intervention, compared to usual care, amongst veterans with PD in the United States [38]. The CHAPS is an outpatient programme, adapted from the Chronic Care Model [39], in which nurse care managers carry out structured six-monthly assessments by telephone, followed by development of a patient-centred care plan, which incorporates evidence-based protocols and coaching in self-management skills. Primary outcome measures were adherence to 38 PD care quality indicators. The eight secondary outcome measures focused on health-related quality of life and perceptions of care quality, assessed via structured telephone survey interviews at 6, 12 and 18 months. Adherence to care quality indicators was significantly better within the intervention group but, with the exception of a reduction in depressive symptoms, this did not appear to translate into other tangible benefits for patients [36].

In summary, since the emergence of the PDNS role in the UK, four trials, spanning several decades and all except one taking place in UK healthcare settings, have evaluated PDNSs as the primary focus.

### 3.3. International Implementation

Since Rosemary Maguire’s appointment in 1989, several countries have embedded PDNSs into practice, based on the UK model. In the United States, there are established educational programmes for nurses wishing to specialise in PD [40], as well as the Edmund J. Safra visiting nurse faculty programme, which trains nurse educators (those who train nursing students) [41]. The first PDNS in Australia started in 1997 as a pilot, modelled on the UK role [42]. In a survey of individuals with PD living in Australia, access to a PDNS was ranked second in importance only to research funding [43]. In May 2008, the Western Australian Department of Health adopted the National Institute for Health and Care Excellence (NICE) 2006 guidelines, which includes provision of PDNSs, as the best practice framework [42,44]. However, a 2011 report stated there were only 33 PDNSs in Australia [45], a country with a population of over 22 million at that time [46]. Comparatively, there were 264 in the UK at that time [45], albeit for a population nearly three times that of Australia. Only 8.4% of 491 people with PD, living in New Zealand and surveyed in 2006–2007, had ever seen a nurse specialist [47]. The issues of access have been further highlighted in a pan-European survey [48].

The World Health Organisation Neurology Atlas 2004, highlighted that neurological nursing was not a recognised specialty in 41% of the 109 countries included [49]. Whilst the PDNS role is established in the developed world, the median number of neurological nurses varies from 0 per 100,000 population for low-income countries, including those in Africa, to 5.04 per 100,000 for higher middle-income countries, including central and eastern Europe [49]. A door-to-door prevalence study of individuals in the Hai region of Tanzania found that 78% of identified cases were previously undiagnosed [50]. Recognising the under-diagnosis of PD in Sub-Saharan Africa [50], lack of access to specialists [51] and limited availability of medication, including Levodopa [52], the Movement Disorder Society (MDS) African Task Force was set up in 2012 [53].

One of the main aims of the MDS African Task Force was to promote awareness of PD and develop training. In December 2012 the first PDNS course was delivered in Tanzania, led by established UK-based PDNSs and attended by nursing professionals, as well as therapists, from six East African countries [54]. This course was delivered in English by an international faculty and covered topics, including diagnosis, management of motor and non-motor symptoms, PD medication and provision of information, as well as featuring the perspectives of local patients with PD [55]. Following the success of this course, others have been run in Ghana (2013), South Africa (2014), Ethiopia (2016), with a further course in Tanzania in 2019 (see Figure 3) [56].

Training PDNSs has the potential to have a significant impact on the care of people with PD in Sub-Saharan Africa, especially given the lack of doctors with movement disorder-specific expertise in these countries [51]. Dotchin et al. described the crucial role of the PDNS in overseeing the up-titration of levodopa in the Tanzanian PD cohort, particularly when initiating medication late in the disease course [58]. Qualitative research undertaken in Tanzania, has demonstrated the lack of awareness around PD and the deeply held views surrounding symptoms, which individuals attribute to witchcraft, a curse or an expected part of the ageing process [59]. This highlights an important role for the PDNS in educating patients, caregivers and the wider community, as well as dispelling cultural-specific myths around the disease. As a result of this pioneering work, participants from more than 20 countries have attended training and have been linked with a UK PDNS mentor and there are ambitions for a network of neurology nurses in Africa. These successful face-to-face training programmes and previous use of tele-education for PD training in Cameroon [60] provide a model with which to facilitate dissemination of the PDNS role to other low-income countries.

## 4. Discussion

Thirty years since the first PDNS was employed in the UK, this professional group are now a central part of the MDT who care for people with PD. We have highlighted their wide-reaching role, which includes providing vital psychological and emotional support to patients and caregivers, delivering education, coordinating care and utilising their specialist skills and expertise in the management of this complex disease.

To date, four randomised controlled trials have, to our knowledge, compared care involving a PDNS to care without PDNS input [20,31,35,36]. However, we acknowledge that this was a narrative review and, therefore, did not follow the full process of a systematic review, which could have resulted in our review omitting relevant literature and could have introduced an element of selection bias. As summarised, studies have so far failed to show a positive impact on clinical outcomes, although patients value their contribution [31,61]. The lack of clear positive outcomes may be a result of various limitations of the trials. The trials conducted by Jahanshahi et al. and Reynolds et al. both had a small sample of patients (*n* = 40 with PD and n = 108 who completed the study, respectively), without explicit mention of power having been calculated for the chosen outcomes [20,31]. Additionally, some patients randomised by Reynolds et al. to receive care from a neurologist only, were referred to see the PDNS during the trial period; these individuals were analysed separately from the consultant-only, PDNS-only and combined care groups, but whilst the change in score from baseline to the end of the study is reported for each dimension for each group, the difference in change between groups is not reported [20]. In the trial by Reynolds et al., patients in the control arm were cared for at sites which had a PDNS, which may have led to contamination [20].

The study populations further limit the generalisability of the findings. Connor et al. recruited almost entirely male veterans (95.7% and 98.8% in the intervention and control arms, respectively) [36], Reynolds et al. recruited only patients newly referred to the clinic [20], thus, likely over-representing people who were early in the disease course. Jahanshahi et al. [31] and Reynolds et al. [20] only included participants with ‘no clinical evidence of dementia’, whilst Jarman et al. [35] and Connor et al. [38] excluded participants with cognitive impairment if they could not provide informed consent, therefore, excluding a group who often have motor fluctuation, a high burden of neuropsychiatric symptoms and complex psychosocial needs, who may potentially have most to gain from PDNS involvement. These trials were all conducted in the UK or USA, which limits their generalisability to other countries, especially given the differences in nomenclature for defining PD nurse roles, which also reflects global differences in the scope of the role and training required.

The nature of the intervention and the duration over which it was implemented also limits the conclusions that can be drawn. Jahanshahi et al. provided a clear description of the PDNS role, along with the frequency of PDNS contact for active arm patients, which is a strength, as it allows the intervention to be replicated if desired. However, the 6-month intervention period may not have been long enough to show benefit [31]. Reynolds et al. reported that participants were reviewed at baseline and 12 months as a minimum and some were seen up to five times during the study, although the authors do not state what proportion of participants received more frequent review and what triggered this [20]. In Jarman et al.’s trial, nurses recorded their activities at work over two 1-week periods [35]. All the nurses were new to community-based PD nursing and undertook a course during the trial period, so there is likely to have been a period of new skill acquisition, which could have attenuated the benefit [37]. In the intervention trialled by Connor et al., PDNSs took a care management approach by conducting structured telephone assessments, with subsequent treatment guided by protocols and patient priorities [38] and the published process evaluation suggests good quality and quantity of intervention implementation [61].

It may be that the true benefit of PDNS involvement is only seen when they are part of a multicomponent intervention, with the goal of fully integrated care. van der Marck et al. evaluated an integrated, multidisciplinary care intervention within a multicentre, non-randomised, controlled trial (IMPACT) in the Netherlands [62]. After their initial routine visit to the neurologist, patients in the intervention group met their local Parkinson’s disease nurse, who offered them a multidisciplinary assessment at the tertiary referral centre [62]. After adjustment for baseline disease severity, there was no difference in quality of life or activities of daily living scores between the intervention and control groups [62]. However, a subsequent RCT in Germany, comparing standard neurological care to an intervention delivered by a ‘PD team’ and consisting of a tailored treatment plan, PD nurse home visits and a telephone hotline, found improved quality of life favouring the intervention [63]. This suggests that an integrated, multidisciplinary approach, with PD nurses as part of the team, may be beneficial. Finally, there is the question of whether suitable outcome measures were used in these trials, since they did not explore effects on caregiver wellbeing or burden, patient activation levels or the degree to which people with PD rate care as patient centred. Forthcoming studies, including the planned Cost-effectiveness of Nursing Interventions for Patients with PD (NICE-PD) study [64], and an RCT of specialised nursing interventions in PD underway in China [65], may be better able to tackle these considerations.

Although evidence to support a positive impact of PDNS on clinical outcomes is lacking, NICE recognises their key role, recommending that PD patients in the UK should have access to a PDNS for support, information and monitoring [66]. The recent biannual UK Parkinson’s Audit showed that 97.8% of patients had access to a PDNS or equivalent, although fully integrated clinics, in which doctors, PDNSs and therapists all see patients in one venue, are only available at 17.7% of sites [67]. Only 88.7% of patients who completed the Patient Reported Experience Measure (PREM) questionnaire reported access to a PDNS, indicating a possible need for education around how to access PDNSs, but this finding should be considered in light of the likely skewed sampling frame towards patients functionally and cognitively able to complete this measure and potential bias towards reporting, where concerns had arisen [67].

PDNSs are valued by patients and caregivers [20,30] and defined by NICE as a quality standard. Yet, recent surveys and reports have raised concerns about gaps in provision and rising caseloads for PDNSs. NICE guidance recommends a caseload of 300 patients per PDNS [68], yet over 80% of respondents to a national questionnaire-based study reported caseloads exceeding this, with a mean and median reported caseload of 526 and 490, respectively [69]. A similar survey in 2011 revealed that 19% of PDNSs have caseloads of over 700 patients [70]. The USP project is currently seeking to understand the scope and value of PDNSs in the UK, in part, to inform the case for commissioning [71]. With rising prevalence of PD [12] and mounting pressures on acute services, PDNSs would appear to be vital members of the team that we can ill-afford to lose.

## 5. Conclusions

The nurse specialist role arose in the 1950s, within the fields of rheumatology and diabetes, and was adopted into Parkinson’s care, with the first PDNS originating in the UK in 1989. Whilst the current evidence base for the PDNS role is limited by several methodological issues, they are broadly regarded as an effective and valuable part of the Parkinson’s MDT. Since the introduction of PDNSs in the UK, the role has been adopted widely, including across Europe, Australia and the USA. The success of the educational programme, delivered by UK-based PDNSs across Sub-Saharan Africa, is an example of how training can be adapted and disseminated, including in lower income countries. Future, well-designed and methodologically robust trials, evaluating the clinical and cost-effectiveness of these roles, will provide a better evidence base to inform healthcare commissioning decisions and leverage further support for the PDNS role. This, in turn, should help to tackle caseload, enhance training, and make the PDNS role fit for the challenges of 21st century medicine.

## Figures and Tables

**Figure 1 geriatrics-07-00046-f001:**
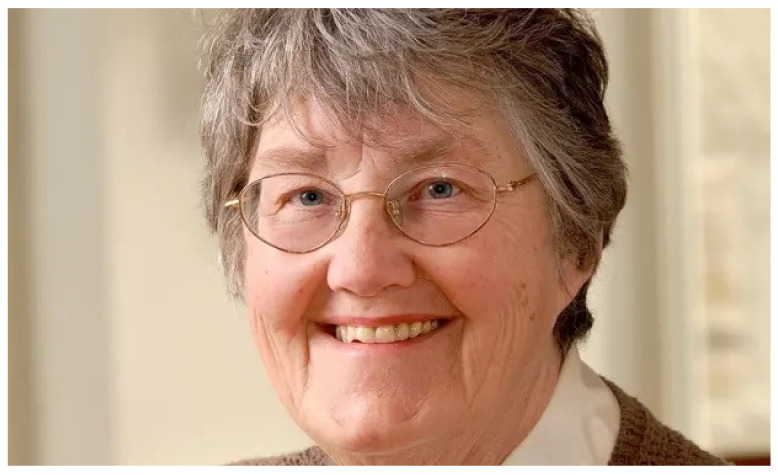
Rosemary Maguire MBE, the first Parkinson’s disease nurse specialist, who was appointed in the UK in 1989. Photo taken by Chris Saville, reprinted with permission from Mike Wilk at Apex News and Pictures.

**Figure 2 geriatrics-07-00046-f002:**
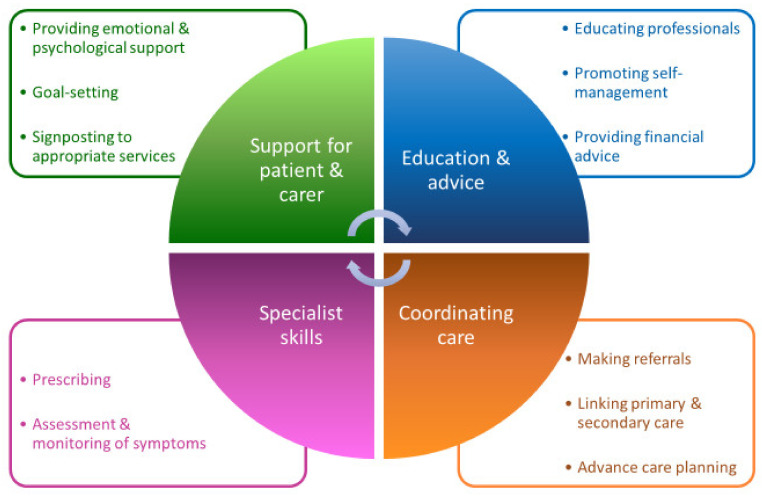
Roles of the Parkinson’s Disease Nurse Specialist. (Data extracted and adapted from [2,24,25,26].)

**Figure 3 geriatrics-07-00046-f003:**
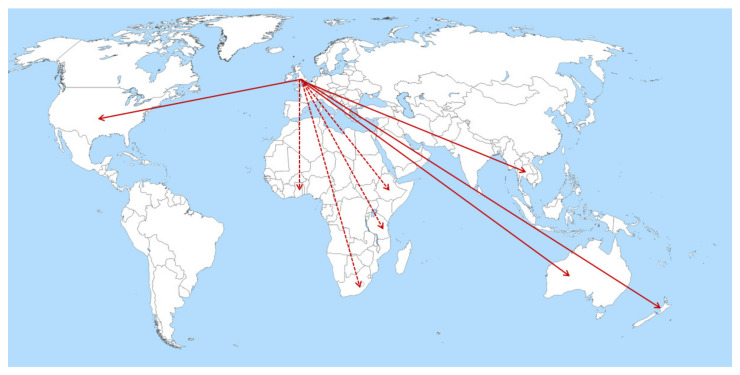
World map displaying the origin of Parkinson’s Disease Nurse Specialists in the UK, with subsequent adoption of a PDNS service globally, including Australia, Thailand [57] and the USA (solid lines). UK-led training programmes have additionally helped to disseminate skills and information to several countries in Africa (dotted lines).

**Table 1 geriatrics-07-00046-t001:** Summary of randomised controlled trials of Parkinson’s Disease Nurse Specialist care. BDI, Beck Depression Inventory; STAI, Spielberger Trait Anxiety Inventory; AIS, Acceptance of Illness Scale; PDNS, Parkinson’s Disease Nurse Specialist; HADS, Hospital Anxiety and Depression Scale; SF-36, 36-item short form survey; PDQ-39, Parkinson’s Disease Questionnaire-39; GP, general practitioner, PHQ-2, Patient Health Questionnaire-2.

Author	Year	Duration (months)	Setting	Study Arms	Participants		Outcome Measure(s)	Key Findings (Intervention Group vs. Control)
					Number Enrolled	Mean Age (yrs)		
Jahanshahi et al. [31]	1994	6	1 tertiary centre in England	Intervention 2. home visits and 5 telephone contacts with a nurse practitioner Control No contact with nurse practitioner (1:1 randomisation)	40 (PD)	63.7	-9 psychosocial measures assessed via a set of self-completed questionnaires (including BDI, STAI, AIS) -7 questions to assess patient satisfaction	7 contacts with a nurse practitioner: -did not appear to positively or negatively impact psychological wellbeing. -was highly valued by patients. -led to a high rate of referrals to other health care professionals
Reynolds et al. [20]	2000	12	3 UK outpatient clinics	Intervention Hospital-based PDNS care only (group B) or predominantly PDNS with Consultant neurologist follow-up (2 to 5 contacts during the study period) (group C) Control (group A) Consultant follow-up care only (Randomisation according to ‘established pattern of care’ at each centre)	185	Group A: 65.1 B: 68.5 C: 67.6	-Health state assessed via 6 questionnaires (HADS, SF-36, PDQ-39, Functional Disability Scale, Patient and Carer Satisfaction Survey, Social requirements), from which 22 dimensions were derived -Healthcare costs	-PDNSs led to similar outcomes but at increased cost -Medical and nursing specialists valued the complementary expertise of PDNSs
Jarman et al. [35]	2002	24	9 regions of England (community-based)	Intervention Community-based care from a nurse specialist who advised GP (8 nurse-led assessments per year on average). Control Standard care from GP (Randomisation within practices with variable randomisation ratio to equate workload)	1859	Approx. 1/3 in each group: 18– 70, 71–77, > 77	Primary outcomes -Objective health measures (stand up test, dot in square score, mortality, proportion sustaining fracture) -Patient wellbeing (PDQ-39, EuroQoL; global subjective wellbeing question) -Healthcare costs Secondary outcomes Medication; Referrals	-Little effect on the objective health measures -Improved sense of wellbeing with no increase in healthcare costs
Connor et al. [36]	2019	24	5 medical centres in the U.S.A.	Intervention A coordinated, nurse-led, chronic care management intervention Control Usual outpatient care (1:1 randomisation at patient level)	328	71.0	Primary outcome: adherence to 38 quality of care indicators Secondary outcomes: Health-related quality of life; perceptions of care quality (structured telephone survey interviews) Healthcare costs	-Improved adherence to PD quality of care indicators -Moderate evidence of a reduction in depressive symptoms (PHQ-2)

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
