# Peer review of "A Narrative Review of Specialist Parkinson’s Nurses: Evolution, Evidence and Expectation"

_geriatrics, 2022, doi:10.3390/geriatrics7020046_

Round 1

Reviewer 1 Report

Thank you very much for this interesting paper, dedicated to a very topical, emerging and important subject.

I would like to start with some general comments and then move into more detail on the several sections of the paper:

In fact, I found that rather outdated references were used throughout all sections of the paper - even though the field of Parkinson's nurse research is a rather small one, there are well-researched publications, especially more recently, that were not included in this paper. This criticism applies to both the narrative review conducted and to all other sections of the paper.

In addition, the list of references shows a great number of irregularities and partly an incorrect citation style - this should be revised. Also, within the text e.g., in line150 the study described at this point was not cited, but another one.

Overall, the list of references and the quotation should be revised both in terms of content and form.

These are my remarks regarding the introduction:

Unfortunately, this comment actually refers back to the references: In line 29,47 and 49 'now' is used, but paper from 2003 and 2009 are referred to. Again, there are more recent papers that could be better cited at this point.

I appreciate the introduction to the history of the PDNS - thank you very much for that. However, I am not quite sure if the personal story of Rosemary Maguire plus photo is relevant for the publication. Here, one could think about removing the photo and the narration from lines 57-62.

In line 66, the authors announce that they will reflect on the three decades of evolving of the PDNS in the paper. Except for the section in the introduction, I have not found any more information on this.

Comments on the methods:

Even though the authors explicitly state that no systematic literature search was performed, I miss a rather detailed description of the methodology.

It is not clear to me why only the two databases listed were searched, how many articles or journal articles were found, who performed the search, and how the quality and consistency of the search and publications found were checked.

Comments on results:

I actually enjoy the headings that lead the reader through the three sections of this part of the paper.

The first few sentences describing Parkinson's in 3.1 would, in my opinion, be more appropriate in the Introduction.

Thank you for the informative graph in this section. Unfortunately, I miss where this originates from and where the information for it was taken from.

In this section, the authors point out the roles that PDNS can play and aim for a global comparison. Unfortunately, they do not address that the designation of Parkinson's disease nurses and the associated training paths vary widely around the world and are not unified. Maybe the authors should strongly point out that the paper has a strong focus on the British training path of the Parkinson Nurse.

I also suggest to take the setting of care delivery into account (in-vs. outpatient and acute care). That also refers to the next section where I also miss the distinction.

In the section on the evidence on effectiveness I wonder why the authors have only found and integrated four RCTs as there are more around on the role of the Parkinson Nurse. Again, latest publications seem not to be included and/or found. If you allow me, I would suggest to have a look at the work of an der Mark or Eggers et al. I think the paper would gain more value in revising more publications here.

Table 1 should be checked and revised for spelling or adherence to a consistent format. Some of the lines are written in bold; capitalization is not consistent. In the table, I also miss an overview of the type of randomization and a description of the care model underlying the included studies. I consider the comparability and generalizability of the studies included here to be poor due to different outcomes, data collection periods and also survey decades (!). Also, the setting in which the studies were conducted (in-outpatient and acute) is not addressed. Here, too, the paper could benefit enormously from the inclusion of more recent scientific findings.

At the end of the section, I would suggest adding a short summary; otherwise, the transfer to the next topic section has little value.

In the section on the global perspective, I miss a proof of the statement of the first sentence (line 182), moreover, in line 188, the point at the end of the sentence must be positioned after the citation.

In terms of content, I miss the inclusion of data from the USA and Canada and other countries (e.g., South America), especially when it comes to describing the 'global perspective'. Here, the one-sided focus on the British PDNS designation and training of the Parkinson's nurse seems to allow only the comparison with Australian data and those collected from Tanzania.

I personally consider this entire third section to be too one-sided in terms of its description, although it could be seen as an example of the dissemination of PDNS from England; unfortunately, however, this section does no justice to the description of a global perspective. I suggest here the revision of 3.3 and the solely focus on a 'case study', e.g., Tanzania.

In the fourth section of the paper, the discussion, I miss that neither the role of the PDNS has been sufficiently described up to this point nor, for example, the individual roles named by the authors have been discussed. For example, how the delivery of Coordinated Care' (line 240) might look like by the PDNS is not mentioned in the paper.

The discussion addresses significant aspects of the results and debunks the statement of the paper in a stringent and comprehensible way, however, the limitations I addressed earlier, such as the setting, the different naming of PDNS, and the inconsistent training and education of PDNS worldwide, should be addressed here as well if one focus of the paper is to take a global perspective and provide generalizable statements.

I suggest removing the last section on the costs of PDNS (lines 306-311), as this subject would qualify as a paper in itself and is not thematically useful to the paper.

All in all, it can be said that this paper attempts to describe the history of the emergence of PDNS, to describe the role of PDNS, and then to provide a comparison of them across several countries. In some parts this is successful, although a focus on the most important issues and main countries would certainly sharpen the paper.

Also, by revising and reviewing more recent literature the paper would gain more weight and certainly focus the issue.

My best greetings to the authors and thank you for your valuable research.

Reviewer 2 Report

Dear authors, another contribution to this important field of Parkinson Nurses. Her article appears as a kind of historical contribution to the subject of Parkinson Nurses. The result of the literature search seems to have been meager, but benevolently one must note, the topic is just coming back into focus. 

Round 2

Reviewer 1 Report

Thanks to the authors for their careful revision of the paper. The paper has gained in both content and clarity. I do not have any further revision requests. 

Author Response

We thank the reviewer for reviewing our revised manuscript so swiftly and for acknowledging the careful revisions we have made.